# Close Contacts, Infected Cases, and the Trends of SARS-CoV-2 Omicron Epidemic in Shenzhen, China

**DOI:** 10.3390/healthcare10112126

**Published:** 2022-10-26

**Authors:** Furong Li, Fengchao Liang, Bin Zhu, Xinxin Han, Shenying Fang, Jie Huang, Xuan Zou, Dongfeng Gu

**Affiliations:** 1School of Public Health and Emergency Management, Southern University of Science and Technology, 1088 Xueyuan Avenue, Fuguang Community, Taoyuan Street, Nanshan District, Shenzhen 518055, China; 2Shenzhen Center for Disease Control and Prevention, Shenzhen 518073, China

**Keywords:** COVID-19, Omicron, China, strategy

## Abstract

(1) The overall trends of the number of daily close contacts and infected cases as well as their association during an epidemic of Omicron Variant of SARS-CoV-2 have been poorly described. (2) Methods: This study was to describe the trends during the epidemic of the Omicron variant of SARS-CoV-2 in Shenzhen, China, including the number of close contacts and infected cases as well as their ratios by days and stages (five stages). (3) Results: A total of 1128 infected cases and 80,288 close contacts were identified in Shenzhen from 13 February 2022 to 1 April 2022. Before the citywide lockdown (14 March), the number of daily close contacts and infected cases gradually increased. However, the numbers showed a decrease after the lockdown was imposed. The ratio of daily close contacts to daily infected cases ranged from 20.2:1 to 63.4:1 and reached the lowest during the lockdown period. The growth rate of daily close contacts was consistent with those of infected cases observed 6 days later to some extent. (4) Conclusions: The Omicron variant epidemic was promptly contained by tracing close contacts and taking subsequent quarantine measures.

## 1. Introduction

In the past few months, cases of the Omicron variant of SARS-CoV-2 have been reported from several regions of China [1,2], posing new challenges for epidemic prevention and control. Shenzhen, an international metropolis bordering Hong Kong, was one of the cities most affected by the COVID-19 epidemic caused by the Omicron variant in the first half of 2022. Thanks to the timely actions taken by the local government in the early stage, the epidemic has been under control since April.

Among nonpharmaceutical interventions (NPIs), tracking and isolating each close contact and providing optimal medical services are the most effective preventive measures in the early stages of the epidemic [3]. Additionally, close contacts are the high-risk population, and the trends of the number of close contacts might help predict the number of future infected cases. However, although some previous studies have reported the epidemiological characteristics and disease incidence among close contacts [4,5,6,7], there are inadequate data about the overall trends of the number of daily close contacts and infected cases and their association during the epidemic in a metropolis.

The present study aimed to investigate the trends of the epidemic in the early stage based on the data of close contacts and infected cases in Shenzhen, China, from 13 February to 1 April 2022, so as to provide a strategical basis for formulating and improving the prevention and control strategies for the SARS-CoV-2 epidemic.

## 2. Methods

### 2.1. Source of Data

Located in the Guangdong province of South China, Shenzhen has more than 20 million migrants who work, live, and study there. The study used data from 1128 infected cases and 80,288 close contacts collected by the Shenzhen Center for Disease Control and Prevention (Shenzhen CDC) between 13 February and 1 April 2022.

### 2.2. Definitions of Infected Cases, Close Contacts, and Medical Observation

Infected cases were confirmed by two repeated positive results from the local CDC by using PCR tests and further confirmed at the designated hospital by another PCR test. An expert group in Shenzhen traced close contacts based on detailed epidemiological investigations and big data technology. In most cases, close contacts were considered as those who had been in close contact with suspected or confirmed cases but had not taken effective protection measures 4 days before the onset of symptoms or before nucleic acid testing of asymptomatic cases.

According to the policy in place during the Omicron epidemic, close contacts should be quarantined for 14 days after the last unprotected or suspicious contact with a suspected or infected case. Those not testing positive at the end of the observation period would have their health monitored at home for 7 days; following this, they could resume normal daily life if not infected.

### 2.3. Statistical Analysis: Analyzing the Progress of the Epidemic Based on the Growth Rate of Close Contacts

In the present study, the definitions of the growth rates of close contacts and infected cases were as follows:(1)Growth rate of close contacts = (number of new close contacts on day t + 1 − number of new close contacts on day t)/number of new close contacts on day t.(2)Growth rate of infected cases = (number of new infected cases on day t + 1 − number of new infected cases on day t)/number of new infected cases on day t.

Since the citywide lockdown would significantly influence the implementation of intervention policies and individual behaviors, we used only the data before the lockdown (i.e., 14 March 2022) for counting close contacts. As such, the analysis of close contacts based on prelockdown data might capture the early transmission of the epidemic and provide an objective estimate of the infectivity and transmission characteristics of the Omicron variant to some extent. As Spearman’s correlation assesses monotonic relationships (whether linear or not), the present study used Spearman’s correlation to estimate the relationship between the growth rate of daily close contact and infected cases.

STATA 14.0 (StataCorp LP, College Station, TX, USA) was used to perform the analyses. Statistical significance was set at *p* < 0.05.

## 3. Results

### 3.1. Number of New Close Contacts and Infected Cases by Days or Stages

In Shenzhen, the epidemic became increasingly widespread after 13 February 2022. On 11 March, the number of daily confirmed cases peaked for the first time at 80; correspondingly, the number of daily close contacts reached 8229, which was the highest during the pandemic. On 15 March, the number of daily confirmed cases reached the second peak of 97, with corresponding close contacts numbering 3167 (Figure 1). Generally, the number of daily close contacts decreased during the lockdown period, while there was an increase in the daily infected cases in the first 2 days after the initiation of the lockdown.

The ratio of the number of daily close contacts to the number of daily infected cases showed an oscillating downward trend before the lockdown but an oscillating upward trend postimplementation of the lockdown (Figure 2). It is important to note that the ratio reached its lowest levels between 14 March and 20 March, ranging from 20.2:1 to 63.4:1, probably because the entire city was under lockdown during this stage and the population mobility was almost “static”.

Based on the results of the nucleic acid test and overall public health strategy, the outbreak of the Omicron epidemic in Shenzhen can be divided into five stages (Table 1). The initial stage was before 15 February, when sporadic cases occurred in the city, and nucleic acid testing was carried out irregularly at this stage; from 16 February to 24 February, a negative COVID-19 report within the last 48 h was required for access to all public places (stage 2). As the epidemic progressed, a negative COVID-19 test within the last 24 h was required from 25 February to 13 March (stage 3). Subsequently, the local government decisively imposed a one-week citywide lockdown (stage 4), during which all public activities were suspended, and all inhabitants were subjected to three rounds of nucleic acid tests. As the lockdown restrictions gradually eased, life in the city gradually returned to normal (stage 5).

Figure 3 shows the ratio of the number of daily close contacts to the number of daily infected cases in each stage. Overall, the ratio continued to decrease until it reached its nadir (36.8:1) during the lockdown stage.

### 3.2. Predicting the Trajectory of the Epidemic Based on the Growth Rate of Close Contacts

To explore whether and to what extent the trends of daily close contacts could help predict the trends of further daily infections, we calculated the Spearman’s coefficient between the growth rate of daily close contacts (before the lockdown) and the growth rate of daily infected cases over a week (i.e., with a lag of 1–7 days). It was seen that the correlation reached the peak with a lag of 6 days, with a corresponding Spearman’s coefficient of 0.400 (Figure 4). Radar figures indicated that the trend of the growth rates of daily close contacts from 14 February to 13 March synchronized most with those of daily infected cases from 20 February to 19 March (lag of 6 days), compared with the analysis of other lag days (Figure 5).

## 4. Discussion

Currently, the Omicron variant of SARS-CoV-2 is the dominant strain worldwide. Major cities in China, such as Shanghai and Jilin, have been hit hard by the outbreak of the epidemic, and strict measures have been taken to prevent the spread of the Omicron variant [8,9]. Shenzhen, one of the most developed port cities in the Guangdong province, effectively controlled the transmission risk within a short period of time when the epidemic broke out in March 2022. Our study reported the trends of daily close contacts and infected cases of the Omicron epidemic in Shenzhen and found that a one-week citywide lockdown appeared to greatly reduce the progression of the epidemic. Meanwhile, the growth rates of close contacts in the early stages were somewhat consistent with those of infected cases that were observed 6 days later.

As the epidemic of SARS-CoV-2 has transformed into sporadic regional outbreaks in China, the government is adhering to the ‘dynamic zero-COVID-19 strategy’ [1,10]. As one of the most important NPIs, contact tracing and subsequent quarantine measures have effectively restricted the spread of the epidemic in China [11,12,13]. Compared with the earlier forgoing of close contact statistics in many countries, China’s use of big data and precise epidemiological investigation to assist in controlling the sources of infection has led to remarkable achievements [14,15], especially in coordination with citywide lockdowns whenever necessary. Generally speaking, although lockdowns and mass testing are not common for most countries at present [16], curbing the COVID-19 epidemic may still have important public health implications in China. A recent study developed a perdition model and concluded that the Omicron variant would cause substantial surges in hospitalizations, ICU admissions, and deaths, and would overwhelm the healthcare system; in fact, the Omicron wave would lead to approximately 1.55 million deaths if the dynamic zero-COVID strategy were rejected [17]. Moreover, besides the acute symptoms related to the infection, the long-term health complications of COVID-19, namely long COVID, is also a big challenge for the public health community [18,19,20].

In the present study, we also found that the ratio of daily close contacts to daily infected cases reached the lowest level during the lockdown period, mainly due to the relatively small number of close contacts under ‘static’ conditions. Overall, the rapid containment of the Omicron epidemic in Shenzhen has largely confirmed the effectiveness of the measures mentioned above in containing the risk of new transmission.

The probability of infection among close contacts of confirmed SARS-CoV-2 cases can be considered an essential indicator to evaluate the transmissibility of the virus. Hence, it is particularly important to study the dynamics of close contacts. Our findings showed that in the early stage of an epidemic episode, the correlation between the growth rates of daily close contacts and those of daily confirmed cases was strongest on the sixth day of the lag, suggesting that the trends of the growth rate of close contacts might predict the trends of confirmed cases 6 days later. This might help to identify and predict future trends in the progression of the SARS-CoV-2 epidemic, and to plan the deployment and storage of treatment resources.

To the best of our knowledge, our study was the first to simultaneously report the trends of daily close contacts and infected cases, as well as the relationship between them, in a metropolis during the Omicron epidemic. Several limitations, however, should be considered. First, close contacts were identified using big data technology (e.g., a mobile base station) and self-reported history; however, there is a probability of omissions or misclassifications. Second, when the number of infected cases reached a certain level, the epidemiological investigation team was unable to undertake extensive fieldwork, which might have led to an inaccurate identification of close contacts. Third, as the lockdown policies are no longer taken by most of the countries and our results were based on data during the epidemic of the Omicron variant of SARS-CoV-2 in a Chinese metropolis, our main findings may not be generalizable to other countries or other scenarios. Fourth, the frequency of nucleic acid testing varied across different epidemic stages, and thus some infected cases may not have been detected in time or may have been missed. Fifth, the sample size was relatively small, the relationship between the growth rate of daily close contact and infected cases may be affected by some extreme values, and the results should be interpreted with caution. Finally, we were not able to have the detailed information on the vaccine rate and the interventions that people were taking (e.g., mask-wearing), both of which might have influenced the infection rate and the dynamics.

## 5. Conclusions

In the context of the ‘dynamic zero COVID-19 strategy,’ tracing close contacts, followed up by subsequent quarantine measures and locking down cities whenever necessary, could be effective measures to swiftly contain the Omicron variant epidemic. Based on the number of daily close contacts, the trend of further progression of the epidemic might be inferred about 6 days in advance.

## Figures and Tables

**Figure 1 healthcare-10-02126-f001:**
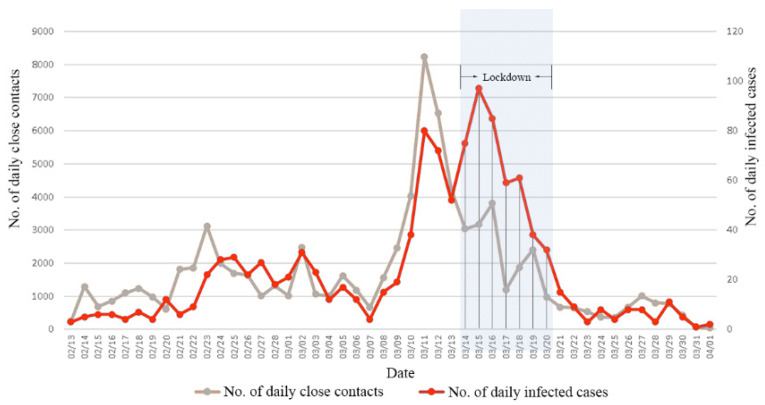
Ratios of the number of daily close contacts and infected cases during the Omicron epidemic in Shenzhen, China, in the spring of 2022. (Note: The grey area indicates the citywide lockdown period.).

**Figure 2 healthcare-10-02126-f002:**
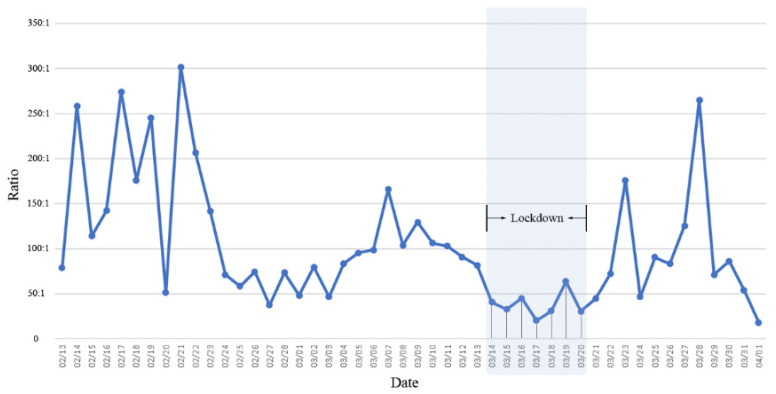
Ratios of the number of daily close contacts and infected cases by different stages during the Omicron epidemic in Shenzhen, China, in the spring of 2022. (Note: The grey area indicates the citywide lockdown period.)

**Figure 3 healthcare-10-02126-f003:**
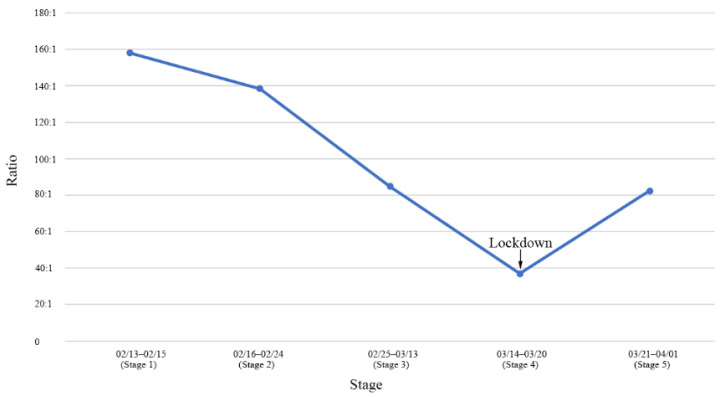
Ratios of the number of daily close contacts and infected cases by different stages during the Omicron epidemic in Shenzhen, China, in the spring of 2022.

**Figure 4 healthcare-10-02126-f004:**
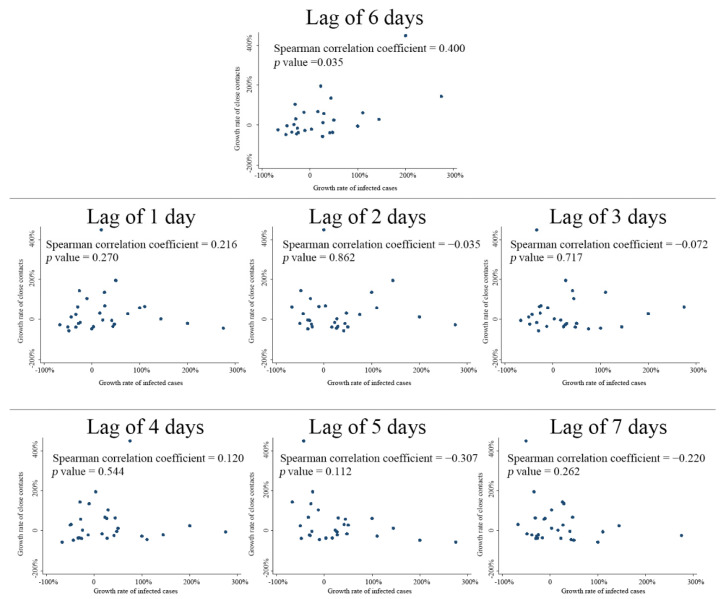
Spearman’s correlation coefficient between the growth rate of daily close contacts (before lockdown) and infected cases by different lag time number of daily close contacts and the number of daily infected cases, by different lag days.

**Figure 5 healthcare-10-02126-f005:**
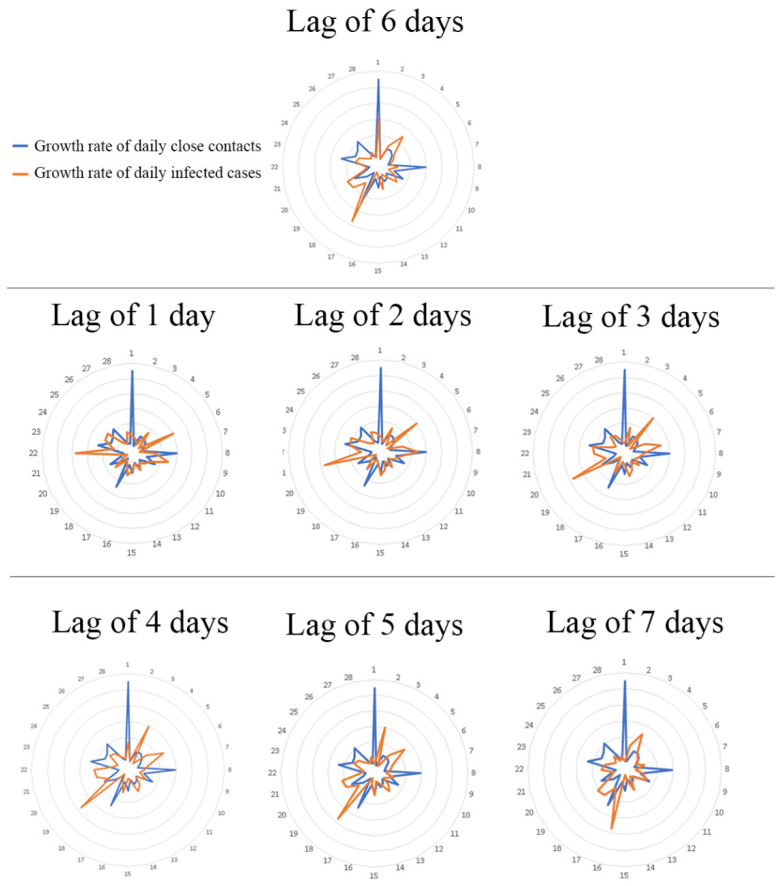
Radar chart of the growth rate of daily close contacts (before lockdown) and infected cases by different lag times. (Note: Each number in the figure corresponds to a single date. For example, in the radar figure for a lag of 6 days, number 1 indicates 2/14 for the growth rate of daily close contacts, whereas it also means 2/20 (lag of 6 days) for the growth rate of infected cases.)

**Table 1 healthcare-10-02126-t001:** The number of daily close contacts and infected cases by different epidemic stages during the Omicron epidemic in Shenzhen, China, in the spring of 2022.

Stage	Date	No. of Daily Infected Cases	No. of Daily Close Contacts
Stage 1	2/13–2/15	14	2212
Stage 2	2/16–2/24	98	13,562
Stage 3	2/25–3/13	492	41,715
Stage 4	3/14–3/20	447	16,458
Stage 5	3/21–4/01	77	6341

Note: The initial stage was before 15 February 2022, when sporadic cases occurred in the city; from 16 February 2022 to 24 February 2022, negative COVID-19 testing within 48 h was required in all public places (stage 2). As the epidemic progressed, negative COVID-19 testing within 24 h was further required from 25 February 2022 to 13 March 2022 (stage 3). Subsequently, the local government decisively imposed a one-week citywide lockdown (stage 4), during which all public activities were suspended and 3 rounds of nucleic acid tests were conducted among all inhabitants. As lockdown restrictions gradually eased, life in the city was gradually returning to normal (stage 5).

## Data Availability

The main dataset used for this study is available upon request.

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
