# Peer review of "Close Contacts, Infected Cases, and the Trends of SARS-CoV-2 Omicron Epidemic in Shenzhen, China"

_healthcare, 2022, doi:10.3390/healthcare10112126_

Round 1
Reviewer 1 Report
Thank you for the opportunity to review your paper. Overall, the issue of SARS-CoV-2 remains a significant issue within healthcare.
Discurssion- I would explicitly the role of public healthcare contibutions realted to contol of SARS-CoV-2 Omicron epidemic to this section.
Author Response
Reviewer #1
The authors thank you for your constructive comment. We have used “Track Changes” to highlight the changes in the revised manuscript. Below is our response to your comment.
Comment 1: Discursion- I would explicitly the role of public healthcare contributions related to control of SARS-CoV-2 Omicron epidemic to this section.
Response: Thank you for this kind guidance. The following public health implications were added to the revised manuscript accordingly: “Generally speaking, although lockdowns and mass testing are not common for most countries at present[1], curbing the COVID-19 epidemics may still have important public health implications in China. A recent study developed a perdition model and concluded that the Omicron variant would cause substantial surges in hospitalizations, ICU admissions and deaths, and would overwhelm the healthcare system; in fact, the Omicron wave may lead to approximately 1.55 million deaths if the dynamic zero-COVID strategy were rejected[2]. Moreover, besides the acute symptoms related to the infection, the long-term health complications of COVID-19, namely long COVID, is also a huge challenge for the public health community[3-5].”
References:
- 1. Han X, Li X, Zhu B, et al. Effect of Lockdown and Mass Testing for the SARS-CoV-2 Omicron Epidemic on Reducing New Infections in Shenzhen, China. Healthcare, 2022;10(9):1725.
- 2. Cai J, Deng X, Yang J, et al. Modeling transmission of SARS-CoV-2 Omicron in China. Nat Med, 2022.
- 3. Koc HC, Xiao J, Liu W, Li Y, Chen G. Long COVID and its Management. Int J Biol Sci, 2022;18(12):4768-80.
- 4. Ceban F, Ling S, Lui L, et al. Fatigue and cognitive impairment in Post-COVID-19 Syndrome: A systematic review and meta-analysis. Brain Behav Immun, 2022;101:93-135.
- 5. Montani D, Savale L, Noel N, et al. Post-acute COVID-19 syndrome. Eur Respir Rev, 2022;31(163).
Reviewer 2 Report
The authors try to describe the association between close contacts and infected cases in Shenzhen, China, from February to April 2022. There are following points that should be addressed before further considerations.
1) What is the definition of the infected cases, whether this is based on single positive test result from PCR test or antigen test, or accompany with any other symptoms?
2) How frequently are the people being tested? This could tie to the potential underestimation of cases.
3) Do you have any explanations of such big difference of pattern when using lag 1 day to lag 6 days? Especially when using lag 5 days show a negative association at the statistical borderline while lag 6 days became significant positive association? Also, we see very flat null results for the remaining lag days, and this needs to be explained and validated.
4) You need to mention the limited generalizability of this study and the potential application as most of the country are not taking any lock-down policies anymore.
5) You also need to mention the vaccine rate and interventions that people are taking (wear mask or not, etc) as they are highly correlated with the infection rate and all the dynamics. If you don't have these information, then you should definitely mention this as a limitation.
Author Response
Reviewer #2
Many thanks to the reviewer for reading the manuscript and giving us the valuable comments. We have used “Track Changes” to highlight the changes in the revised manuscript. Below are our point-by-point responses to your questions and comments.
Comment 1: What is the definition of the infected cases, whether this is based on single positive test result from PCR test or antigen test, or accompany with any other symptoms?
Response: Thank you for this comment. The presence of COVID-19 was confirmed by two repeated positive results from the local Chinese Centers for Disease Control and Prevention (CDC) by using PCR tests; all the infected cases were re-confirmed by another PCR test at The Third People's Hospital of Shenzhen, which is the designated hospital for the admission of patients with COVID-19 in Shenzhen. Based on this comment, the following definition was added to the revised manuscript: “Infected cases were confirmed by two repeated positive results from the local CDC by using PCR tests and further confirmed at the designated hospital by another PCR test.”
Comment 2: How frequently are the people being tested? This could tie to the potential underestimation of cases.
Response: Thank you for this thoughtful comment. In general, the frequency of the test varied during the Omicron wave. In the Results section, the relevant information was provided: “The initial stage was before February 15th, when sporadic cases occurred in the city, and nucleic acid testing was carried out irregularly at this stage; from February 16th to February 24th, a negative COVID-19 report within the last 48 h was required for access to all public places (stage 2). As the epidemic progressed, a negative COVID-19 test within the last 24 h was required from February 25th to March 13th (stage 3). Subsequently, the local government decisively imposed a one-week city-wide lockdown (stage 4), during which all public activities were suspended and all inhabitants were subjected to 3 rounds of nucleic acid tests.”
Being reminded by this comment, the following limitation was added to the revised manuscript: “Fourth, the frequency of nucleic acid testing varied across different epidemic stages, and some infected cases may not be detected in time or may have been missed.”
Comment 3: Do you have any explanations of such big difference of pattern when using lag 1 day to lag 6 days? Especially when using lag 5 days show a negative association at the statistical borderline while lag 6 days became significant positive association? Also, we see very flat null results for the remaining lag days, and this needs to be explained and validated.
Response: Thank you for these comments. The contradictory results could partly result from some extreme values in a relatively small sample size, which may lead to exaggeration or underestimation of the correlation. For example, we can see a pair of extreme values in the top left-hand corner in the figure describing the correlation of lag-5 day; similarly, the extreme values appear in the top right-hand corner in the figure describing the correlation of lag-6 days.
Notably, being reminded by another reviewer, we have used Spearman correlation rather than the Pearson correlation in the revised manuscript, as Spearman correlation can deal with the data better. Furthermore, to avoid misunderstanding, the fitted lines were removed from the figures, as these lines may mislead the readers into thinking that the relationship is linear. In this revised version, the Spearman’s coefficient for lag 5 days and lag 6 days were -0.307 (P value =0.112) and 0.400 (P value =0.035), respectively (Figure 4 in the revised manuscript). Once again, the seemly contradictory results may be due to the extreme values; After removing the extreme values, the corresponding Spearman’s coefficients are respectively -0.150 (P value =0.209) and 0.332 (P value =0.057), which seems to be attenuated after the removal.
Based on this comment, the following limitation was added to the revised manuscript: “Fifth, the sample size was relatively small, and the relationship between the growth rate of daily close contact and infected cases may be affected by some extreme values, thus the results should be interpreted with caution.” Furthermore, to make a more conservative conclusion, the following sentence was removed from the Abstract section: “Based on the growth rate of close contacts, the future trend of the epidemic may be inferred about 6 days in advance.”
Comment 4: You need to mention the limited generalizability of this study and the potential application as most of the country are not taking any lock-down policies anymore.
Response: Thank you for this kind guidance. The following limitation was added to the revised manuscript accordingly: “Third, as the lockdown policies are no longer taken by most of the countries and our results were based on data from the epidemic of the Omicron variant in a Chinese metropolis, our main findings may not be generalizable to other countries or other scenarios.”
Comment 5: You also need to mention the vaccine rate and interventions that people are taking (wear mask or not, etc) as they are highly correlated with the infection rate and all the dynamics. If you don't have these information, then you should definitely mention this as a limitation.
Response: Thank you for this kind guidance. Unfortunately, we do not have the abovementioned information. Therefore, the following limitation was added to the revised manuscript accordingly: “Finally, we were not able to have the detailed information on the vaccine rate and the interventions that people were taking (e.g., mask-wearing), both of which might have influenced the infection rate and the dynamics”.
Reviewer 3 Report
Comments
This study aimed to investigate the trends of the epidemic in the early stage based on the data of close contacts and infected cases in Shenzhen, China, from February 13th to April 1st 2022. In general, the structure and contents are well organized with solid scientific foundation, yet there are still some issue need to be solved/improved before acceptance.
1. In the abstract, the aim of the study should be mentioned in the background.
2. In the results, Pearson correlation and radar charts were applied with figures, but the analyses were not mentioned in the method part. The authors need to describe all the statistical methods they used in the methods, and explain the reasons of choosing the methods, e.g., choosing Pearson correlation but not Spearman correlation in this study. What statistical software (name and version) the authors used to analyze the data? It should be mentioned except for using Microsoft Office.
3. In figure 4, the trend is decreasing in lag-5 days, which is opposite to lag-6 days. How to explain the contradictory results. P-value=0.059 in figure 5 indicates a marginally significant statistical association, even when α<0.05 is regarded as the significance level. And this should also be indicated and discussed.
4. There are some language and grammatic errors in the manuscript and need to be revised.
Author Response
Reviewer #3
The authors thank you for your constructive comments. We have used “Track Changes” to highlight the changes in the revised manuscript. Below are our point-by-point responses to your questions and comments.
Comment 1: In the abstract, the aim of the study should be mentioned in the background.
Response: Thank you for this comment. The beginning of the abstract was changed to “Objective: To describe the overall trends of the daily close contacts and infected cases counting and their association during the epidemic of Omicron Variant of SARS-CoV-2.”
Comment 2: In the results, Pearson correlation and radar charts were applied with figures, but the analyses were not mentioned in the method part. The authors need to describe all the statistical methods they used in the methods, and explain the reasons of choosing the methods, e.g., choosing Pearson correlation but not Spearman correlation in this study. What statistical software (name and version) the authors used to analyze the data? It should be mentioned except for using Microsoft Office.
Response: We apologize for the missing information mentioned above. Being reminded by this comment, we decided to use Spearman correlation rather than the Pearson correlation in the revised manuscript.
Spearman's correlation is a nonparametric measure of rank correlation (statistical dependence between the rankings of two variables). It assesses how well the relationship between two variables can be described using a monotonic function. In fact, while Pearson's correlation assesses linear relationships, Spearman's correlation assesses monotonic relationships (whether linear or not). For the present study, the relationship between the growth rate of daily close contact and infected cases by different lag time may not be linear, we therefore used Spearman's correlation to describe the relationship.
Based on this comment, the following sentence was added to the revised manuscript accordingly: "As Spearman's correlation assesses monotonic relationships (whether linear or not), the present study used Spearman's correlation to estimate the relationship between the growth rate of daily close contact and infected cases. STATA 14.0 (StataCorp LP, College Station, TX) was used to perform the analyses. Statistical significance was set at P < 0.05."
Comment 3: In figure 4, the trend is decreasing in lag-5 days, which is opposite to lag-6 days. How to explain the contradictory results. P-value=0.059 in figure 5 indicates a marginally significant statistical association, even when α<0.05 is regarded as the significance level. And this should also be indicated and discussed.
Response: Thank you for this thoughtful comment. The contradictory results could partly result from extreme values in a relatively small sample size, which may lead to exaggeration or underestimation of the correlation. For example, we can see a pair of extreme values in the top left-hand corner in the figure describing the correlation of lag-5 day; similarly, a pair of extreme values appears in the top right-hand corner in the figure describing the correlation of lag-6 days.
As mentioned in the response to comment 2, we have used Spearman’s correlation rather than the Pearson’s correlation in the revised manuscript. Furthermore, to avoid misunderstanding, the fitted lines were removed from the figures, as these lines may mislead the readers into thinking that the relationship is linear. In this revised version, the Spearman’s coefficient for lag 5 days and lag 6 days were -0.307 (P value =0.112) and 0.400 (P value =0.035), respectively (Figure 4 in the revised manuscript). Once again, the seemly contradictory results may be due to the extreme values, although the P value for lag 5 days is not statistically significant (P value =0.112); After removing the extreme values, the corresponding Spearman’s coefficients are respectively -0.150 (P value =0.209) and 0.332 (P value =0.057), which seems to be attenuated after the removal.
Based on this comment, the following limitation was added to the revised manuscript: “Fifth, the sample size was relatively small, and the relationship between the growth rate of daily close contact and infected cases may be affected by some extreme values, thus the results should be interpreted with caution.” Furthermore, to make a more conservative conclusion, the following sentence was removed from the Abstract section: “Based on the growth rate of close contacts, the future trend of the epidemic may be inferred about 6 days in advance.”
Comment 4: There are some language and grammatic errors in the manuscript and need to be revised.
Response: Thank you for your comment. We have checked the whole manuscript carefully and corrected any language and grammatic errors as far as possible.
Round 2
Reviewer 2 Report
Thanks for the revision and responses.
Reviewer 3 Report
The authors havr replied the comments from the reviewer and the manuscript can be accepted after proofreading.